# Pathogenic Impacts of Dysregulated Polycomb Repressive Complex Function in Hematological Malignancies

**DOI:** 10.3390/ijms22010074

**Published:** 2020-12-23

**Authors:** Satoshi Kaito, Atsushi Iwama

**Affiliations:** Division of Stem Cell and Molecular Medicine, Center for Stem Cell Biology and Regenerative Medicine, The Institute of Medical Science, The University of Tokyo, 4-6-1, Shirokanedai, Minato-ku, Tokyo 108-8639, Japan; kaito@ims.u-tokyo.ac.jp

**Keywords:** polycomb repressive complexes, hematologic malignancies, acute leukemia, myelodysplastic syndromes, myeloproliferative disease, synthetic lethality

## Abstract

Polycomb repressive complexes (PRCs) are epigenetic regulators that mediate repressive histone modifications. PRCs play a pivotal role in the maintenance of hematopoietic stem cells through repression of target genes involved in cell proliferation and differentiation. Next-generation sequencing technologies have revealed that various hematologic malignancies harbor mutations in PRC2 genes, such as *EZH2*, *EED*, and *SUZ12*, and PRC1.1 genes, such as *BCOR* and *BCORL1*. Except for the activating *EZH2* mutations detected in lymphoma, most of these mutations compromise PRC function and are frequently associated with resistance to chemotherapeutic agents and poor prognosis. Recent studies have shown that mutations in PRC genes are druggable targets. Several PRC2 inhibitors, including EZH2-specific inhibitors and EZH1 and EZH2 dual inhibitors have shown therapeutic efficacy for tumors with and without activating *EZH2* mutations. Moreover, *EZH2* loss-of-function mutations appear to be attractive therapeutic targets for implementing the concept of synthetic lethality. Further understanding of the epigenetic dysregulation associated with PRCs in hematological malignancies should improve treatment outcomes.

## 1. Introduction

Polycomb repressive complexes (PRCs) are composed of polycomb group (PcG) proteins and are epigenetic regulators that mediate repressive histone modifications [1]. There are two major types of PRCs, PRC1 and PRC2. Canonical PRC1 consists of polycomb group ring finger (PCGF) 4/BMI1 or PCGF2/MEL18, RING1A or RING1B, CBX, and PHC. RING1A/B is a ubiquitin ligase (E3 ligase), and PRC1 monoubiquitinates histone H2A at lysine residue 119 (H2AK119ub1). PRC2 mainly consists of EZH1/2, EED, SUZ12, and RBBP4/7. EZH1/2 has specific histone methylation activity and mediates the mono-, di-, and trimethylation of H3 at lysine 27 (H3K27me1/me2/me3) (Figure 1) [2]. Generally, PRC2 is recruited to target gene loci, which typically contains non-methylated CpG islands. Then, PRC1 binds to H3K27me3 through one of its components, CBX, and monoubiquitinates H2AK119. These histone modifications inhibit transcription and induce chromatin aggregation, which maintains transcriptional repression of target genes. Recently, several variant PRC1 complexes were identified that can monoubiquitinate H2AK119 independent of PRC2 or H3K27me3. Among them, PRC 1.1, comprising PCGF1, RING1A/B, BCOR or BCORL1, and KDM2B, is of particular interest because of its distinct functions in hematopoietic stem cells and hematologic malignancies [3].

PRCs have been shown to function in many aspects of hematopoiesis, and next-generation sequencing has revealed gain-of-function and loss-of-function mutations in PRCs and related genes. In this review, we present the current knowledge on PRC aberrations in hematologic malignancies and their prognostic significance, as well as potential therapeutic approaches targeting PRCs.

## 2. Physiological Functions of PRCs in Hematopoiesis

Among the various PRC components, BMI1 (also known as PCGF4) has been extensively investigated in hematopoiesis. In Bmi1 deficient mice, *Cdkn2a* (*Ink4a/Arf*), which encodes Ink4a and Arf tumor suppressors, was de-repressed [4]. As a result, *Bmi1* knockout mice showed a significantly lower frequency of hematopoietic stem cells (HSCs) due to impaired self-renewal [5]. In addition, the forced expression of Bmi1 enhanced the self-renewal of HSCs [6]. Bmi1 also suppresses the commitment and differentiation of HSCs into B cells by repressing *Ebf1* and *Pax5*, which encode key transcription factors that are essential for B-cell differentiation. Consequently, *Bmi1*-deficient mice showed premature expression of *Ebf1* and *Pax5* in HSCs and multipotent progenitors (MPPs), which was accompanied by accelerated lymphoid specification and a marked reduction in HSC/MPPs [7]. Bmi1 is also required to maintain leukemic stem cells as well as normal HSCs [8].

The functions of EZH1 and EZH2 in hematopoiesis have also been well characterized. *Ezh2*-deficient HSCs retained their reconstitution capacity [9] because Ezh1 can compensate for Ezh2 [10,11]. Conditional deletion of *Ezh1* in adult mice impaired the self-renewal capacity of HSCs through de-repression of *Cdkn2a* [12], while HSCs in *Ezh1*-null mice retained almost normal repopulation capacity. *Ezh1* and *Ezh2* double knockout mice completely lost repopulation capacity [10]. With respect to other PRC2 components, loss of Eed impaired differentiation of HSCs and led to HSC exhaustion [13]. Suz12 has also been shown to be required for the maintenance of HSC [14].

Several studies have revealed the physiological roles of PRC1.1 in hematopoiesis. KDM2B binds to nonmethylated CpG islands through its zinc finger-CxxC (ZF-CxxC) DNA-binding domain, thereby recruiting other components of PRC1.1. *Kdm2b*-deficient mice had significantly decreased numbers of hematopoietic stem and progenitor cells (HSPCs) [15] which showed impaired repopulation of hematopoietic cells, particularly lymphoid cells [15,16]. In contrast, forced expression of *Kdm2b* prevented exhaustion of the long-term repopulating potential of HSCs following serial transplantation [17]. BCOR, a co-repressor of BCL6, played an important role in restricting differentiation toward the myeloid lineage, partly by repressing *Cebp* and *HoxA* family genes. As a consequence, *Bcor* knockout mice showed myeloid-skewed differentiation [18,19]. Depletion of PCGF1 also led to myeloid-skewing [20] and de-repressed expression of *HoxA* family genes [21]. Taken altogether, these data show that PRC1.1 regulates the functions of HSCs and restricts their differentiation toward the myeloid lineage by repressing the transcription of genes required for myeloid differentiation, such as *C/EBP* and *HOXA* family genes.

## 3. Functions of PRCs in Hematologic Malignancies

Dysregulated function of epigenetic regulators is frequently involved in the pathogenesis of solid and hematological malignancies. PRCs play a pivotal role in the maintenance of HSCs and hematopoiesis, and dysregulation of PRC function has been implicated in the pathogenesis of hematological malignancies. Overexpression of PcG genes generally promotes tumorigenesis, partly through their ability to transcriptionally repress tumor suppressor genes, such as the *CDKN2A* locus (encoding p16^INK4A^ and p14^ARF^), and developmental regulator genes [22]. The expression of *BMI1/PCGF4* has correlated with disease progression and the prognosis of myelodysplastic syndrome (MDS) [23], the prognoses of acute myeloid leukemia (AML) and chronic myeloid leukemia [24]. In contrast, loss-of-function mutations in PRC genes, such as *EZH2*, *EED*, *SUZ12*, *BCOR*, and *BCORL1*, have been found in various hematological malignancies, suggesting their tumor suppressor functions (Table 1) [25,26,27,28,29,30,31,32,33,34,35,36,37,38,39,40,41,42,43,44,45,46,47,48,49,50,51]. Therefore, PcG genes may have dual roles as oncogenes and tumor suppressor genes, depending on the situation and underlying disease. Among the observed PcG gene mutations, genetic abnormalities in *EZH2* and *BCOR* are of particular interest owing to their relatively high frequencies, pathological significance, and potential as therapeutic targets.

### 3.1. EZH2

*EZH2* is frequently overexpressed and/or amplified in prostate, breast, bladder, and colon cancers [52], and its expression is correlated with metastasis [53] and poor prognosis [54,55]. We and other groups have shown that Ezh2 exerts an oncogenic function during the maintenance phase of MLL-AF9 AML in mice and could be therapeutically targeted. In contrast, Ezh2 acts as a tumor suppressor during the induction phase of AML [56,57,58]. EZH2 is strongly expressed in germinal center (GC) B cells and acts with BCL6 to recruit a noncanonical PRC1-BCOR complex containing CBX8 in a GC B-cell–specific manner to repress the expression of differentiation genes [59]. Correspondingly, gain-of-function mutations in *EZH2* are frequently found in GC B-cell–type lymphoma [47] in which H3K27me3 levels are significantly elevated [47]. Mutant EZH2 contributes to lymphomagenesis partly by repressing *Prdm1* and/or *Irf4*, which are essential for B-cell differentiation [60,61,62]. Mutant EZH2 not only increases the abundance of H3K27me3 but also causes the widespread redistribution of this repressive mark, suggesting that mutant EZH2 induces persistent repression and aberrant activation of EZH2 target genes [62].

Loss-of-function mutations in *EZH2* are frequently found in patients with MDS and myeloproliferative neoplasms (MPN). Although *EZH2* mutations are rare in de novo AML patients, they are frequently found in patients with secondary AML transformed from MDS [31]. Abnormalities of chromosome 7, including -7 and -7q, are frequently found in patients with MDS, and they involve *EZH2*, which is located in the long arm of chromosome 7 (7q36). In patients with MDS, *EZH2* mutations are associated with poor prognosis when compared to that of MDS patients with wild-type *EZH2*. Furthermore, patients with bi-allelic *EZH2* mutations showed worse overall survival than those with mono-allelic *EZH2* mutations [47]. In MDS and AML, *EZH2* expression is also regulated by mutations in spliceosome genes, such as *SRSF2* and *U2AF1*, which cause mis-splicing of *EZH2*, leading to degradation of variant *EZH2* mRNAs via nonsense-mediated decay [63,64]. *EZH2* mutations frequently co-occur with *TET2*, *RUNX1*, and *ASXL1* mutations [63]. We have demonstrated that loss of Ezh2 cooperated with a *Tet2* hypomorph or a *RUNX1* mutant to induce MDS and MDS/MPN in mice [10,65,66]. The deletion of *Ezh2* in mice conferred a growth advantage to HSCs and promoted myeloid-biased repopulation. The deletion of *Ezh2* resulted in an MDS/MPN-like disease with myeloproliferative features such as the enhanced repopulating capacity of HSCs and extramedullary hematopoiesis in the spleen, and myelodysplastic features such as anemia accompanied by enhanced apoptosis in erythroblasts and dysplasia in myeloid cells. Concurrent depletion of *Ezh2* and *Tet2* established more advanced myelodysplasia and markedly accelerated the development of myelodysplastic disorders, including both MDS and MDS/MPN [9,57].

These findings clearly indicate that EZH2 plays a tumor-suppressive role in myelodysplastic disorders and enhances transformation triggered by other driver mutations. In combination with the *RUNX1S291fs* mutant, loss of Ezh2 also promoted development of MDS in mice. In this MDS model, we found that *RUNX1S291fs/Ezh2*-null cells compromise normal HSPCs function [65,66]. In MDS patients, mutant cells propagate in the bone marrow (BM) despite impaired proliferative capacity and are thought to eventually outcompete normal hematopoietic cells. *RUNX1S291fs/Ezh2*-null MDS cells appeared to be defective in proliferative capacity. Of interest, inflammatory cytokine pathways such as IL-6 pathway, which are inhibitory towards HSC functions were significantly activated in residing normal HSPCs, but not in *RUNX1S291fs/Ezh2*-null HSPCs. Expression of IL-6 was significantly elevated in *RUNX1S291fs/Ezh2*-null BM cells compared with Rx291 BM cells. Excessive IL-6 production by MDS cells has also been shown to promote leukopenia and thrombocytosis in a 5q- MDS mouse model [67]. These findings suggest that *RUNX1S291fs/Ezh2*-null cells outcompete normal cells by creating an inflammatory BM environment, thereby achieving clonal expansion despite their compromised proliferative capacity.

Essential tumor suppressor and developmental regulator genes remain transcriptionally repressed even in *EZH2*-insufficient cells [10,65]. HSPCs are maintained in mice with only one *Ezh1* allele (that lack *Ezh2*), while deletion of both Ezh1 and Ezh2 leads to the depletion of HSPCs. Residual PRC2 in mice with only one *Ezh1* allele preferentially targeted genes with high levels of H3K27me3 and H2AK119 monoubiquitination (H2AK119ub1) in HSPCs. We designated these genes as Ezh1 core target genes. Most Ezh1 core target genes were bivalent developmental regulators. These results indicate that Ezh1 targets bivalent genes to maintain self-renewing stem cells in *EZH2*-insufficient MDS [11].

*Ezh2* insufficiency in mouse HSPCs leads to a switch from H3K27me3 to acetylation (H3K27ac) at promoter regions of many PRC2 target genes, which is closely associated with increased transcription of a subset of direct polycomb targets, including those with bivalent promoters [56,65,66,68,69,70,71]. The absence of Ezh2 induces activation of a cohort of fetal-specific genes, including let-7 target genes [71]. Many of the fetal-specific let-7 target genes, including *Lin28*, *Hmga2*, and *Igfbp3*, are known as oncofetal genes, and are targeted by Ezh2-mediated H3K27me3 in adult BM HSPCs. *Ezh2* insufficiency also activates the production of inflammatory cytokines and proteins, such as interleukin-6 (IL-6), S100a8, and S100a9. In mouse models, a loss of Ezh2 induces a feto-oncogenic program that includes genes such as Plag1, whose overexpression phenocopies a loss of Ezh2 to accelerate induction of AML [56].

Loss-of-function mutations in *EZH2* and *SUZ12* also have been found in early T-cell precursor (ETP) acute lymphoblastic leukemia (ETP-ALL) and non-ETP T-ALL in children and T-ALL in adults (Table 1). Experiments in several mouse models have demonstrated that *Ezh2*-deficient mice develop T-ALL [10,72]. Of note, oncogenic *NOTCH1* mutations are frequently found in T-ALL and induce the loss of H3K27me3 modifications by antagonizing the functions of PRC2, thus activating the transcription of NOTCH1 targets [39]. We also generated an ETP-ALL mouse model by deleting *Ezh2* in p53-null hematopoietic cells. In this ETP-ALL model, a large portion of the PRC2 target genes acquired DNA hypermethylation at their promoters following the loss of Ezh2, which included pivotal T cell differentiation–regulating genes. Thus, PRC2 protects key T cell developmental regulators from DNA hypermethylation to keep them primed for activation during subsequent differentiation phases, while PRC2 insufficiency predisposes ETPs to leukemic transformation [73]. In another mouse model of ETP-ALL, inactivation of PRC2 also caused derepression of *Il6ra*, resulting in activation of JAK/STAT signaling [74]. Mutations in *EZH2* frequently co-occur with inactivating mutations in *RUNX1* in ETP-ALL. Combined deletion of *Ezh2* and *Runx1* in ETPs induced their expansion. The addition of a RAS-signaling pathway mutation (Flt3-ITD) resulted in an aggressive ETP-like leukemia co-expressing myeloid and lymphoid genes [75].

In some cases, mutations in genes encoding histone H3K27 reduce levels of H3K27me3, even in the absence of loss-of-function *EZH2* mutations, leading to tumor development. These mutations, initially found in pediatric brain tumors, cause amino acid substitution such as K to I/M and induce a global reduction in H3K27me3 levels by binding to EZH2 and strongly trapping PRC2, despite the fact that only one copy of the 30-copy histone H3 gene has the mutation [76]. Recently, it was reported that there are cases with these mutations in AML as well, although the frequency of H3K27I/M mutations has been found very low, at 4 out of 434 patients (0.9%) [77]. Furthermore, in AML mouse models, H3K27I/M mutations have been reported to be a potent disease promoter in the *RUNX1-RUNX1T1* AML mouse model [78].

### 3.2. BCOR and BCORL1

Two of the noncanonical PRC1.1 genes, *BCOR* and its homolog *BCORL1*, are both X-linked and are frequently mutated in patients with various hematological malignancies (Table 1). These are inactivating mutations, and they often co-occur with mutations in *DNMT3A*, *TET2*, *RUNX1*, and *STAG2* [34,36,79]. We demonstrated that a *Bcor* mutant that lacks the PCGF1-binding domain combines with Tet2 loss to induce lethal MDS in mice. These MDS cells reproduced MDS or evolved into lethal MDS/myeloproliferative neoplasms in secondary recipients. These findings indicate that BCOR mutations promote the initiation and progression of MDS in concert with concurrent driver mutations [19]. Transcriptional profiling revealed the de-repression of myeloid regulator genes such as *Cebp* and *HoxA* family genes, in HSPCs. The loss of BCOR induced reductions in H2AK119ub1 levels at many promoter regions of PRC1 targets, including *Hoxa7*, *Hoxa9*, and *Cebpa* [19]. This *Bcor* mutant also combines with an oncogenic *Kras* mutation, *Kras^G12D^*, to initiate a fully transplantable acute leukemia [80]. These findings indicate that *BCOR* mutations promote the initiation and progression of MDS and acute leukemia in concert with concurrent driver mutations, indicating its function as a tumor suppressor. Of interest, *Bcor*-insufficient MDS cells do not show activation of Ezh2 target genes, such as let-7 target genes and inflammatory genes described above [19]. These results suggest that PRC1.1 insufficiency is involved in the pathogenesis of MDS in a manner that differs from PRC2 insufficiency (Figure 2).

Although inactivating mutations in *BCOR* and *BCORL1* are not frequent in T-ALL, mice with both *Bcor* insufficiency and *Kdm2b* insufficiency show a strong propensity to develop T-ALL, mostly in a Notch-dependent manner [16,19,81]. BCOR targets many NOTCH1 targets, including Myc, and antagonizes their transcriptional activation [16,81]. A *Bcor* mutant lacking the PCGF1-binding domain also promoted B-cell lymphomagenesis in Eμ-Myc mice [82].

As described above, BCOR also functions as a corepressor of BCL6 and contributes to the formation of a noncanonical PRC1/BCOR complex containing CBX8 in GC B cells, thereby linking BCL6 to EZH2-containing PRC2. The interaction between BCL6-noncanonical PRC1 and BCOR-PRC2 led to the silencing of B-cell differentiation and cell cycle checkpoint genes to permit immunoglobulin affinity maturation. Thus, excessive gene silencing by this machinery is involved in lymphomagenesis [59]. PRC1.1 component genes are significantly overexpressed in primary AML CD34^+^ cells, and downmodulation of these components strongly reduced cell proliferation in vitro and attenuated leukemogenesis in humanized xenograft mouse models [83]. Transgenic mice overexpressing Kdm2b developed myeloid or B-lymphoid leukemia [84]. These findings suggest that noncanonical PRC1.1 also has a context-specific oncogenic function.

### 3.3. ASXL1

*ASXL1* is one of the homologs of the additional sex comb gene, which functions as a chromatin-binding scaffold protein for epigenetic regulators. ASXL1 regulates polycomb functions and trithorax in Drosophila [85]. The loss of ASXL1 results in global reductions in H3K27me3 levels, suggesting that ASXL1 regulates the recruitment of PRC2 to their target loci [86,87]. ASXL1 also forms a complex with the deubiquitination enzyme BAP1, and removes monoubiquitin from H2AK119Ub, to de-repress genes targeted by PRC1 [88]. *ASXL1* is targeted by deletions or somatic mutations in patients with hematologic malignancies, including MDS (14–23%), AML (5–17%), CMML (40–49%), and primary myelofibrosis (13–32%) [89]. *ASXL1* mutations are associated with poor prognosis in many hematologic tumor patients [90,91,92,93]. Recent studies using mice expressing an *ASXL1* mutant, which generated a carboxy-terminal truncated ASXL1 protein, demonstrated that an *ASXL1* mutation alone was not sufficient for inducing the development of hematologic malignancies, but increased the susceptibility to leukemogenesis in concert with a *RUNX1* mutant or in viral insertional mutagenesis, indicating that mice expressing an *ASXL1* mutant represent a premalignant condition [94].

## 4. PRC as a Therapeutic Target

As mentioned above, both loss-of-function and gain-of-function mutations in *EZH2* are found in hematological malignancies. Although these mutations are thought to be involved in tumor development in various ways, a comprehensive understanding is lacking, and treatment challenges remain. Several PRC2 inhibitors targeting the oncogenic function of EZH2 have been developed and are currently in preclinical and clinical studies.

### 4.1. EZH2 Inhibitors

Several EZH2-specific inhibitors have been developed [95] and their safety and efficacy have been evaluated in clinical trials (Table 2) [96,97,98,99]. Except for two clinical trials (NCT03456726, NCT03213665), most of them included patients without *EZH2* mutations. Tazemetostat is the most widely studied EZH2-specific inhibitor, and it was the first to be approved by the FDA (2020) for advanced epithelioid sarcoma. The efficacy of tazemetostat for the treatment of malignant lymphoma has been validated in phase I and II studies [96,97], and a phase II study of tazemetostat showed high efficacy for follicular lymphoma with both wild-type and mutant *EZH2* mutations [97]. In this phase II study, the objective response rate was 69% (31 of 45 patients) in the cohort with *EZH2* mutation and 35% (19 of 54 patients) in those with wild type *EZH2*. Serious treatment-related adverse events were reported only in 4 out of 99 patients (4%). As tazemetostat was well tolerated and the frequency of serious side effects was low, it is a promising treatment for relapsed and refractory follicular lymphoma.

The efficacy of tazemetostat has also been tested in combination with other drugs. For instance, attempts have been made to add tazemetostat to RCHOP (rituximab, cyclophosphamide, doxorubicin, vincristine, and prednisolone), which is the standard of care for diffuse large B-cell lymphoma (DLBCL) (NCT02889523) [98]. EZH2 inhibitors and prednisolone were shown to have synergistic anti-tumor activity in a DLBCL mouse model [100]. In addition, the combination of EZH2 inhibitors with immune checkpoint inhibitors appeared to be effective for the treatment of solid tumor-bearing mice. Mechanistically, it has been proposed that EZH2-mediated gene silencing of chemokines, such as CXCL9 and CXCL10, prevents effector T-cell trafficking to the tumor microenvironment, thereby allowing tumor cells to evade immunity [101]. In light of these basic research results, attempts have been made to combine EZH2 inhibitors with immune checkpoint inhibitors, such as antibodies against PD-L1 or CTLA-4, in the treatment of solid tumors (NCT04407741, NCT03525795). This type of combination therapy may be applicable to the treatment of hematological malignancies.

Inhibitors targeting both EZH1 and EZH2 (EZH1/2 dual inhibitors) and EED (EED inhibitor) have also been developed [102,103,104], and these inhibitors show better tumor-eradicating effects than EZH2-specific inhibitors in the treatment of AML, multiple myeloma, and EZH2 inhibitor-resistant DLBCL cells [105,106,107,108,109,110]. EED stabilizes and activates PRC2 to maintain H3K27 methylation for gene silencing. EED inhibitors are also attracting attention as a potentially promising agent as well as EZH2 inhibitors. One of the EED inhibitors, MAK-683, is currently under clinical trial for the patients with advanced malignancies including DLBCL (NCT02900651).

### 4.2. Hypomethylating Agents

Hypomethylating agents such as azacitidine and decitabine are clinically used in the treatment of high-risk MDS. One of the indications of the hypomethylating agent for MDS treatment is that MDS cells have higher levels of DNA hyper-methylation than *de novo* AML cells [111]. Azacitidine has been shown to improve the overall survival of patients with high-risk MDS [112]. However, it is not always the case that the mutations of epigenetic modifiers that affect the levels of DNA methylation such as *TET2*, *IDH1/2*, and *DNMT3A* predict the response to the hypomethylating agents [113]. Although MDS harboring *TP53* mutations have been reported to temporarily respond better to hypomethylating agent [113], there are few reports that it leads to improved survival. The prediction of response to hypomethylating agents and which cases will obtain the benefits most remain to be elucidated.

The efficacy of the hypomethylating agents for hematologic malignancies harboring *EZH2* mutations also remains unclear due to the relatively small number of cases. However, it has been well documented that PRC2 target genes, particularly bivalent genes marked by H3K4me3 and H3K27me3, defined in embryonic stem cells as well as hematopoietic stem and progenitor cells preferentially display DNA hyper-methylation in tumor cells, such as colon cancer, AML, and MDS, regardless of genetic mutations in epigenetic modifiers [114,115]. In addition, a functional link between DNA methylation and PRC2 has been implicated in the regulation of the biological processes of development, differentiation, and cancer [116,117]. We showed that an epigenetic switch from H3K27me3 to DNA methylation occurs at a significant portion of PRC2 targets following the loss of *Ezh2* in *Tet2* hypomorph (*Tet2^KD/KD^Ezh2^Δ/Δ^*) mice, thereby maintaining their transcriptional repression to promote the progression of MDS [118]. Although Ezh2 and Ezh1 targets both underwent DNA hypermethylation in a similar manner, Ezh2 targets gained higher levels of DNA methylation than those of Ezh1 targets, indicating that Ezh2 targets are the major targets of the epigenetic switch in *Tet2^KD/KD^Ezh2^Δ/Δ^* MDS cells. The gene silencing of Ezh1 targets was regulated mainly by sustained H3K27me3 and only partly by DNA methylation. An epigenetic switch from H3K27me3 to DNA methylation was also evident in the ETP-ALL mouse model established by Ezh2 loss in p53-null hematopoietic cells [73]. In these mice, a large portion of PRC2 target genes acquired DNA hypermethylation of their promoters following reductions in H3K27me3 levels upon the loss of Ezh2. The epigenetic switch was accentuated at the promoter regions of pivotal transcriptional regulator genes in *Ezh2/p53*-deficient ETP-ALL cells, the silencing of which impedes the proper ensemble of transcription factors for T cell development. Of note, the treatment of *Ezh2/p53*-deficient ETP-ALL cells with decitabine resulted in the terminal differentiation of ETP-ALL cells following the reactivation of a set of genes encoding transcriptional regulators of T cell development. These results indicate that the DNA methylation–mediated inhibition of the gene networks of T cell development was largely responsible for the differentiation block of ETPs during the development of ETP-ALL. Thus, PRC2 protects key T cell developmental regulators from DNA hypermethylation in order to keep them primed for activation upon subsequent differentiation phases, while its insufficiency predisposes ETPs to leukemic transformation. These results revealed a previously unrecognized epigenetic switch in response to PRC2 dysfunction and provide the basis for specific rational epigenetic therapy for ETP-ALL with PRC2 insufficiency.

### 4.3. Novel Strategies for Loss-of-Function Mutations in PRC Genes—Synthetic Lethality

Loss of EZH2 has been shown to lead to chemotherapy resistance in both AML [119] and T-ALL [120]. A number of clinical studies have also shown that loss-of-function mutations in *EZH2* are correlated with poor patient outcomes and require specific treatments [121,122,123]. Although drug discovery against loss-of-function mutations has generally been more difficult than drug discovery against gain-of-function mutations, the concept of synthetic lethality, in which cell death results from inhibiting a more dependent pathway, for loss-of-function mutations has recently received much attention. The most well-known genes involved in synthetic lethality are *BRCA* and *PARP* [124], and the usefulness of PARP inhibitors for BRCA-deficient breast and ovarian cancers has been demonstrated in clinical trials [125,126]. Vigorous attempts have been made to find synthetic lethal partners for loss-of-function mutations in *EZH2*.

As described above, PRC2 insufficiency induces leads to a switch from H3K27me3 to acetylation (H3K27ac) at promoter regions of many PRC2 target genes, which is closely associated with increased transcription of a subset of direct polycomb targets, including potential oncogenes in cancer cells. This epigenetic switch may bring about an oncogenic addiction to the H3K27ac modification in *EZH2* insufficient tumor cells. NF1 encodes a RasGTPase-activating protein (RasGAP) and its loss drives cancer by activating Ras. *SUZ12* loss potentiates the effects of NF1 mutations in NF1-associated cancers by amplifying Ras-driven transcription through enhancing acetylation at H3K27 [127]. Bromodomain inhibitors inhibit the function of enhancers by competitively interfering with the binding of BRD4 to H3K27ac and abrogate the progression of tumors [128]. NF1-associated cancers appeared to be sensitive to bromodomain inhibition, indicating that PRC2 insufficiency might trigger an epigenetic switch that sensitizes these cancers to bromodomain inhibitors. We also found that loss of Ezh2 in JAK2V617F mice promotes an epigenetic switch at promoter regions of PRC2 target genes. This results in the activation of potential oncogenes such as *Hmga2*, thereby promoting the development of JAK2V617F-induced myelofibrosis in mice [68]. Loss of PRC2 increases sensitivity to bromodomain inhibition of JAK2V617F myelofibrosis initiating cells in vitro and in vivo [68]. Furthermore, a combination of bromodomain and JAK kinase inhibition reduces NF-kB-induced inflammation, which completely reverses fibrosis in JAK2V617F model mice [129]. These studies suggest that bromodomain inhibitors alone and the combination of bromodomain inhibitor and JAK2 inhibitor can be a novel therapeutic strategy for eradicating hematologic malignancies harboring *EZH2* mutation [127,130].

There are some other promising synthetic lethal partner candidates for EZH2 that were recently discovered. Gu et al. showed that BCAT1, which catalyzes the reversible transamination of branched-chain amino acids, is aberrantly activated in *EZH2*-deficient myeloid neoplasms, and inhibition of BCAT1 selectively impairs *EZH2*-deficient cells. These findings indicate that hematologic malignancies harboring *EZH2* loss-of-function mutations have an increased dependence on BCAT1, making BCAT1 a potential therapeutic target [131]. Leon et al. demonstrated that *EZH2* loss-of-function mutations have increased sensitivity to CHK1 inhibitors caused by enhanced replication stress due to increased expression of MYC-N in T-ALL cells [132].

## 5. Conclusions

With the recent development of next-generation sequencing, it has become clear that many hematological tumors harbor mutations in PRC genes. However, in many cases, the mechanisms by which these mutations contribute to tumorigenesis remain unclear. Tumors with mutations in PRC genes often have poor treatment outcomes, and treatment challenges remain. EZH2 inhibitors targeting PRC2 gain-of-function have been developed, and new therapeutic targets, such as loss-of-function mutations in PRC2 genes, are being actively explored by employing the concept of synthetic lethality. These developments should improve the treatment outcomes of patients with hematological malignancies harboring mutations in PRC genes.

## Figures and Tables

**Figure 1 ijms-22-00074-f001:**
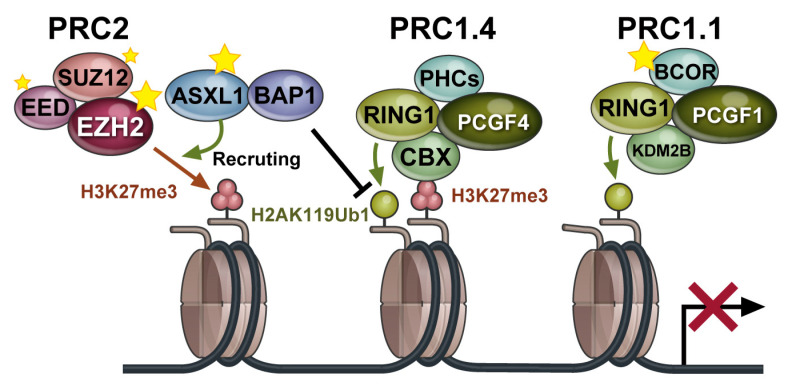
Polycomb repressive complexes (PRCs) are multiprotein complexes that play important roles in the repression of target genes through chromatin modifications. PRC components that are targeted by somatic gene mutations are indicated by stars. Black T arrow indicates inhibition of ubiquitination at H2K119.

**Figure 2 ijms-22-00074-f002:**
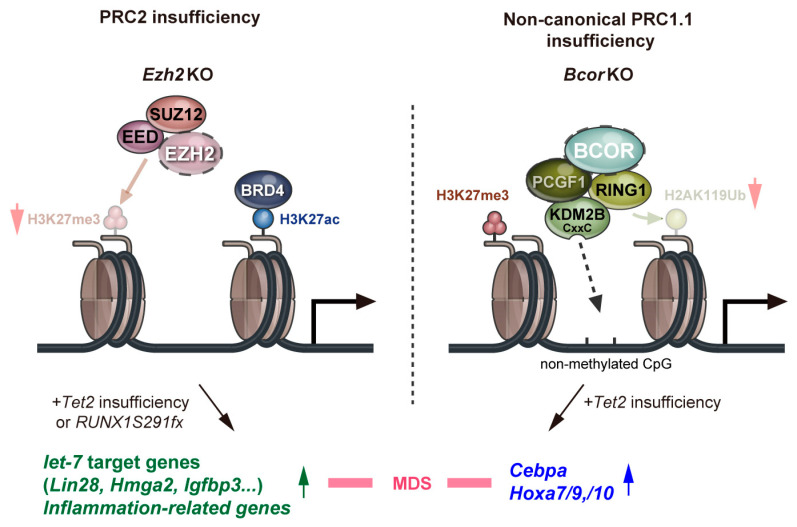
A model for roles of polycomb repressive complexes in the pathogenesis of MDS. Red arrows indicate a decrease in the levels of H3K27 trimethylation and H2K119 ubiquitination. Green and blue arrows indicate increased expression levels of the respective genes.

**Table 1 ijms-22-00074-t001:** Frequencies of mutations in PRC genes in hematologic malignancies.

Diseases			PRC2 Genes			PRC1.1 Genes		Reference
			*EZH2*	*EED*	*SUZ12*	*BCOR*	*BCORL1*	
Myeloid neoplasms	Leukemia	AML	1.5%			2.8–5.0%	3.7–5.8%	[25,26,27,28]
		DS-AMKL	32.7%		2.0%	4.1%		[29]
		secondary AML	5.7–8.6%			7.5–14.3%		[30,31]
	MDS		5.5–6.4%			4.0–5.0%	0.8–0.9%	[30,32,33,34,35,36]
	MPN	PV	3.3%		3.0%			[35,37]
		MF	13.3%					[35]
	MDS/MPN	CMML	10.0–12.7%		3.0%	2.9%		[30,35,38]
		aCML	12.9%	1.8%				[35,38]
		MDS/MPN-U	9.7%					[35]
Lymphoid neoplasms	Leukemia	T-ALL	5.6–16.2%	3.7%	4.4–5.6%			[39,40]
		ETP-ALL	15.6–16.1%	6.5–12.5%	9.7–17.2%			[41,42]
		Non-ETP-ALL	4.8%	7.1%	4.8%			[41]
		T-PLL	9.3%			7.8%		[43,44]
		CLL				2.2%		[45]
	Lymphoma	FL	7.2–27.5% *					[46,47,48]
		DLBCL	9.7–14.3% *					[47,49]
		ENKL				20.6–32.0%		[50,51]

AML, acute myeloid leukemia; DS-AMKL, acute megakaryoblastic leukemia associated with Down syndrome; MDS, myelodysplastic syndromes; MPN, myeloproliferative neoplasms; PV, polycythemia vera; MF, myelofibrosis; MDS/MPN, myelodysplastic/myeloproliferative neoplasms; CMML, chronic myelomonocytic leukemia; aCML, atypical chronic myeloid leukemia; MDS/MPN-U, myelodysplastic/myeloproliferative neoplasms, unclassifiable; T-ALL, T-cell acute lymphoblastic leukemia; ETP-ALL, early T-cell precursor acute lymphoblastic leukemia; T-PLL, T-cell prolymphocytic leukemia; CLL, chronic lymphocytic leukemia; FL, follicular lymphoma; DLBCL, diffuse large B-cell lymphoma; ENKL extranodal NK/T-cell lymphoma, nasal type. * Gain-of-function mutation.

**Table 2 ijms-22-00074-t002:** Clinical trials of PRC2 inhibitors for hematologic malignancies.

	Drugs	Targeted Diseases	Phase	Estimated No. of Participants	Ages Eligible for Study	Identifier	Reference
EZH2 inhibitor	Tazemetostat	B-cell Lymphomas, solid tumors	I/II	420	Adults (20 years or older)	NCT01897571	[96,97]
	Tazemetostat	R/R B-cell NHL harboring *EZH2* mutations	II	13	Adults (20 years or older)	NCT03456726	
	Tazemetostat	R/R NHL, solid tumors harboring *EZH2*, *SMARCB1*, or *SMARCA4* mutations	II	49	Children and Adults (1–21 years)	NCT03213665	
	Tazemetostat	NHL, Rhabdoid tumors, Solid tumors	II	300	Adults (18 years or older)	NCT02875548	
	Tazemetostat Atezolizumab Obinutuzumab	R/R FL, DLBCL	I	96	Adults (18 years or older)	NCT02220842	
	Tazemetostat R-CHOP	DLBCL	I/II	133	Adults (60–80 years)	NCT02889523	[98]
	Tazemetostat Lenalidomide Rituximab	R/R FL	III	518	Adults (18 years or older)	NCT04224493	
	GSK2816126	R/R NHL, MM, Solid tumors	I	41	Adults (18 years or older)	NCT02082977	[99]
	CPI-1205	B-cell Lymphomas	I	41	Adults (18 years or older)	NCT02395601	
	SHR2554	R/R mature lymphoid neoplasms	I	42	Adults (18–70 years)	NCT03603951	
	PF-06821497	FL, DLBCL, Solid tumors	I	172	Adults (18 years or older)	NCT03460977	
	Valemetostat	ATLL	II	25	Adults (20 years or older)	NCT04102150	
EZH1/2 dual inhibitor	HH2853	NHL, solid tumors	I	30	Adults (18 years or older)	NCT04390737	
EED inhibitor	MAK-683	DLBCL, solid tumors	I/II	203	Adults (18 years or older)	NCT02900651	

R/R, relapsed or refractory; NHL, non-Hodgkin lymphoma; FL, follicular lymphoma; DLBCL, diffuse large B-cell lymphoma; R-CHOP, rituximab, cyclophosphamide, doxorubicin, vincristine, and prednisolone; MM, multiple myeloma; ATLL, adult T-cell leukemia/lymphoma.

## Data Availability

No new data were created or analyzed in this study. Data sharing is not applicable to this article.

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
