# Peer review of "Pathogenic Impacts of Dysregulated Polycomb Repressive Complex Function in Hematological Malignancies"

_ijms, 2020, doi:10.3390/ijms22010074_

Round 1
Reviewer 1 Report
This review article provides a summary of the roles and functions of polycomb group proteins in haematological malignancies and comments on their potential as therapeutic targets. Although the authors stated “polycomb repressive complex functions” in the title, I found that the content is heavily biased on the EZH2, with a little bit focus on PRC1.1 components BCOR and BCORL1. The authors may consider elaborate the discussion of other PRC components in hematopoiesis and haematological malignancies, or consider revising the title to be more specific to EZH2 (or PRC2 complex). Besides, the discussion in section 4 about therapeutic targets is a bit vague. The author should elaborate more on the last paragraph of “EZH inhibitors”, and include the discussion of other PcG protein inhibitors (either in experimental or clinical trials) in this section. In addition, while it is interesting to comment on the idea of applying synthetic lethality for the loss-of-function mutations in EZH2, I found that the conclusion of “PRC2 insufficiency triggered an epigenetic switch that sensitizes these cancers to bromodomain inhibitors” (line 317-318) is not well justified, based on the indirect evidence from the loss of SUZ12 in NF1-associated cancers and the epigenetic effects of bromodomain inhibitors on H3K27ac (line 312-316). Lastly, the authors comment on the potential of loss-of-function mutations in EZH2 as therapeutic targets using the examples of increased sensitivity to EZH2 inhibitor in cancer cells with ARID1A or BRG1 mutations (line 338-341). These examples seem not unequivocally representing the situation of “loss-of-function mutations in EZH2”. Perhaps the authors should clarify their argument or cite better examples to support their arguments.
Minor comment:
- Line 287, are the ETP-ALL cells deficient of Ezh2/p53?
Author Response
Dear Reviewer #1,
We are grateful for your important comments and useful suggestions. According to your constructive comments, we believe that the manuscript is now largely improved. As indicated in the responses that follow, we considered all your comments and suggestions in preparing the revised version of our manuscript. All the revised points are highlighted in yellow.
Major comment
This review article provides a summary of the roles and functions of polycomb group proteins in haematological malignancies and comments on their potential as therapeutic targets. Although the authors stated “polycomb repressive complex functions” in the title, I found that the content is heavily biased on the EZH2, with a little bit focus on PRC1.1 components BCOR and BCORL1. The authors may consider elaborate the discussion of other PRC components in hematopoiesis and haematological malignancies, or consider revising the title to be more specific to EZH2 (or PRC2 complex).
Response
We really appreciate your important comment. As pointed out, this review is largely biased on the function of PRC2, because PRC2 is the major target found to be deregulated in hematological malignancies. However, we also referred to the deregulation of PRC1.1, PRC1.4, and ASXL1. Therefore, we would like to keep the title as it is.
Besides, the discussion in section 4 about therapeutic targets is a bit vague. The author should elaborate more on the last paragraph of “EZH inhibitors”, and include the discussion of other PcG protein inhibitors (either in experimental or clinical trials) in this section.
Response
Thank you for your constructive suggestion.
In this revised manuscript, we added several results of recent clinical trials on the efficacies of EZH2 inhibitors in detail.
We added the sentences on page 8, lines 260-262, as follows.
“In this phase 2 study, the objective response rate was 69% (31 of 45 patients) in the cohort with EZH2 mutation and 35% (19 of 54 patients) in those with wild type EZH2. Serious treatment-related adverse events were reported only in 4 out of 99 patients (4%).”
Furthermore, we added some descriptions of EED inhibitors on page 11, lines 284-287, as follows.
“EED stabilizes and activates PRC2 to maintain H3K27 methylation for gene silencing. EED inhibitors are also attracting attention as a potentially promising agent as well as EZH2 inhibitors. One of the EED inhibitors MAK-683 is currently under clinical trial for the patients with advanced malignancies including DLBCL (NCT02900651).”
In addition, while it is interesting to comment on the idea of applying synthetic lethality for the loss-of-function mutations in EZH2, I found that the conclusion of “PRC2 insufficiency triggered an epigenetic switch that sensitizes these cancers to bromodomain inhibitors” (line 317-318) is not well justified, based on the indirect evidence from the loss of SUZ12 in NF1-associated cancers and the epigenetic effects of bromodomain inhibitors on H3K27ac (line 312-316).
Response
We really appreciate your important comment. The reviewer’s point is quite reasonable.
In the revised manuscript, we changed the sentence on page 12, lines 351-352, as follows.
“suggesting indicating that PRC2 insufficiency triggered might trigger an epigenetic switch that sensitizes these cancers to bromodomain inhibitors.”
Lastly, the authors comment on the potential of loss-of-function mutations in EZH2 as therapeutic targets using the examples of increased sensitivity to EZH2 inhibitor in cancer cells with ARID1A or BRG1 mutations (line 338-341). These examples seem not unequivocally representing the situation of “loss-of-function mutations in EZH2”. Perhaps the authors should clarify their argument or cite better examples to support their arguments.
Response
Thank you for your suggestion. We agree with the reviewer’s comment.
We deleted the following part in which we referred to these examples.
“In addition, in solid tumors, it has been shown that EZH2 and the SWI/SNF complex have a synthetic lethal relationship. The SWI/SNF complex plays an important role in chromatin remodeling and is known to act antagonistically with PRC2. ARID1A is a DNA binding component of the SWI/SNF complex that is frequently mutated in ovarian clear cell carcinoma. Ovarian tumors with ARID1A mutations have been shown to be more sensitive to EZH2 inhibitors [111]. Furthermore, lung cancer cell lines harboring mutations in BRG1 (SMARCA4A), which encodes an ATPase of the SWI/SNF complex, are more sensitive to EZH2 inhibitors [112]. These findings indicate that loss-of-function mutations in EZH2 may be a potential therapeutic target, and further research on the relationship between PRC and the SWI/SNF complex in hematology is warranted.”
Minor comment
Line 287, are the ETP-ALL cells deficient of Ezh2/p53?
Response
Thank you for your inquiry. Yes, these ETP are deficient of Ezh2/p53.
To clarify, we changed the sentence on page 12, lines 320-323, as follows.
“Of note, the treatment of Ezh2/p53-deficient ETP-ALL cells with decitabine resulted in the terminal differentiation of ETP-ALL cells following the reactivation of a set of genes encoding transcriptional regulators of T cell development.”
Reviewer 2 Report
The manuscript "Pathogenic impacts of dysregulated polycomb repressive complex function in hematological malignancies" by Satoshi Kaito and Atsushi Iwama is a well written, comprehensive, and timely review on the topic. I have some comments, including additional information to be included to expand and add to the discussion, as well as minor corrections to the text.
- Oncogenic mutations in histone H3 that prevent it's PRC2-mediated methylation are clearly relevant to the discussion. Although these were originally discovered in pediatric brain tumours, more recently their role in hematologic malignancies has been reported. This should be included in the discussion, including the following and possibly other studies on the same topic: https://www.ncbi.nlm.nih.gov/pmc/articles/PMC6599207/; https://ashpublications.org/blood/article/130/20/2204/115023/H3K27M-I-mutations-promote-context-dependent.
- Additional sex combs-like (ASXL) proteins are polycomb group proteins, important regulators of PRC complex activity, and also highly important tumour suppressors mutated in myeloid malignancies in human. They are not currently discussed in the manuscript, and I recommend that they are included at least briefly. There is a large and recent literature to draw on, including molecular biology, human genetics and patient studies, and mouse models.
- The discussion of the molecular mechanisms linking PRC1 dysregulation to malignant transformation is much more limited in the review, as compared to PRC2. I assume this reflects our deeper knowledge of the mechanisms of action and the transcriptional targets of PRC2, as compared to PRC1, in normal and malignant hematopoietic cells. There has been ongoing debate in the literature over the importance of the E3 Ub ligase catalytic activity of PRC1/Ring1B for its activity as a repressor of gene expression. Do we know whether the disruption of PRC1 activity, for example in cancers with BCOR mutations, causes transformations through loss or re-distribution of H2AK119ub epigenetic mark, or is it not directly linked to dysreguation in H2AK119ub, or do we not have these answers yet?
- Do we know whether the regulation/repression of inflammatory genes by PRC2 discussed in the manuscript, including IL6, IL6Ra, S100As, etc, is a direct effect? Do we know if PRC2/Ezh2 is recruited to the regulatory regions of these genes, and Ezh2 loss directly results in loss/depletion of H3K27me3 at the bound site, leading to increased gene expression?
- Section on clinical trials of Ezh2 Inhibitors: Could you specify if the inhibitors show efficacy against epithelioid sarcoma and follicular lymphoma in general, or are the patients in these trials specifically pre-selected to include only those with Ezh2 gain-of-function mutations?
- One of the early studies reporting the role of BMI1 in the maintenance not only of normal HSCs but also of leukemic stem cells in mouse models can be further discussed (PMID: 12714970 DOI: 10.1038/nature01572).
Minor questions and corrections:
- Line 77, "Depletion of PCGF1 also led to myeloid-skewing [19] and repressed expression of HoxA family genes". Do you mean de-repressed? I believe hyperactivation of HoxA genes is usually linked to myeloid transformation.
- Lines 85-90, Two sentences here that are highly informative but lack citations. "Overexpression of PcG genes generally promotes tumorigenesis, partly through their ability to transcriptionally repress tumor suppressor genes, such as the CDKN2A locus (encoding p16INK4A and p14ARF), and developmental regulator genes. The expression of BMI1/PCGF4 has been correlated with disease progression and the prognosis of myelodysplastic syndrome (MDS) as well as the prognoses of acute myeloid leukemia (AML) and chronic myeloid leukemia." Please add some citations.
- Similarly, Line 103 "EZH2 is overexpressed and/or amplified in prostate, breast, bladder, and colon cancers, and its expression is correlated with metastasis and poor prognosis." This is a very general statement, and some citations should be added. Also, can we have more specific information, how frequently EZH2 is dysregulated in these types of cancers? In majority of cases, frequently, occasionally? The sentence in its current form can be misinterpreted to mean that EZH2 activation is always seen in these cancers.
- Line 149, "genes are remain", typo to be corrected.
- The text on lines 160-165 is somewhat repetitive. It seems the same or similar information is stated several times. "The absence of the polycomb-group gene Ezh2 induces activation of a cohort of fetal-specific genes, including let-7 target genes, in adult BM HSPCs, leading to acquisition of fetal phenotypes by BM HSPCs, such as enhanced self-renewal activity and production of fetal-type lymphocytes [63]. Many of the fetal-specific let-7 target genes, including Lin28, Hmga2, and Igfbp3, are known as oncofetal genes, and are targeted by Ezh2-mediated H3K27me3 in adult BM HSPCs. Ezh2 loss results in their ectopic expression in adult BM HSPCs."
Author Response
Dear Reviewer #2,
We are grateful for your critical comments and useful suggestions that have helped us to improve our paper. As indicated in the responses that follow, we have taken all your comments and suggestions into account when preparing the revised version of our manuscript. All the revised points are highlighted in yellow.
Major comment
The manuscript "Pathogenic impacts of dysregulated polycomb repressive complex function in hematological malignancies" by Satoshi Kaito and Atsushi Iwama is a well written, comprehensive, and timely review on the topic. I have some comments, including additional information to be included to expand and add to the discussion, as well as minor corrections to the text.
Response
We really appreciate these encouraging comments and believe that the manuscript was largely improved by the suggested revisions.
- Oncogenic mutations in histone H3 that prevent it's PRC2-mediated methylation are clearly relevant to the discussion. Although these were originally discovered in pediatric brain tumours, more recently their role in hematologic malignancies has been reported. This should be included in the discussion, including the following and possibly other studies on the same topic: https://www.ncbi.nlm.nih.gov/pmc/articles/PMC6599207/; https://ashpublications.org/blood/article/130/20/2204/115023/H3K27M-I-mutations-promote-context-dependent.
Response
Thank you for your essential comment on the oncogenic mutations in histone H3.
In the revised manuscript, we discussed this point on page 6, lines 185-193, as follows with the citations that are kindly recommended from the reviewer.
In some cases, mutations in genes encoding histone H3K27 reduce levels of H3K27me3, even in the absence of loss-of-function EZH2 mutations, leading to tumor development. These mutations, originally found in pediatric brain tumors, cause amino acid substitution such as K to I/M and induce a global reduction in H3K27me3 levels by binding to EZH2 and strongly trapping PRC2, despite the fact that only one copy of the 30-copy histone H3 gene has the mutation [76]. Recently, it has been reported that there are cases with these mutations in AML as well, although the frequency of H3K27I/M mutations has been found very low, at 4 out of 434 patients (0.9%) [77]. Furthermore, in AML mouse models, H3K27I/M mutations have been reported to be a potent disease promoter in the RUNX1-RUNX1T1 AML mouse model [78].
- Additional sex combs-like (ASXL) proteins are polycomb group proteins, important regulators of PRC complex activity, and also highly important tumour suppressors mutated in myeloid malignancies in human. They are not currently discussed in the manuscript, and I recommend that they are included at least briefly. There is a large and recent literature to draw on, including molecular biology, human genetics and patient studies, and mouse models.
Response
We really appreciated your important advice.
As pointed out, ASXL proteins are polycomb group proteins and important regulators of PRC complex activity. In this revised manuscript, we added the sentences regarding ASXL1 in hematologic malignancies on pages 7-8, lines 230-244, as follows.
ASXL1 is one of the homologs of the additional sex comb gene, which functions as a chromatin binding scaffold protein for epigenetic regulators. ASXL1 regulates polycomb functions and trithorax in Drosophila [85]. The loss of ASXL1 results in global reductions in H3K27me3 levels, suggesting that ASXL1 regulates recruitment of PRC2 to their target loci [86, 87]. ASXL1 also forms a complex with the deubiquitination enzyme BAP1, and removes monoubiquitin from H2AK119Ub, to de-repress genes targeted by PRC1 [88]. ASXL1 is targeted by deletions or somatic mutations in patients with hematologic malignancies, including MDS (14-23%), AML (5-17%), CMML (40-49%), and primary myelofibrosis (13-32%) [89]. ASXL1 mutations are associated with poor prognosis in many hematologic tumor patients [90-93]. Recent studies using mice expressing an ASXL1 mutant, which generated a carboxy-terminal truncated ASXL1 protein, demonstrated that an ASXL1 mutation alone was not sufficient for inducing the development of hematologic malignancies, but increased the susceptibility to leukemogenesis in concert with a RUNX1 mutant or in viral insertional mutagenesis, indicating that mice expressing an ASXL1 mutant represent a premalignant condition [94].
- The discussion of the molecular mechanisms linking PRC1 dysregulation to malignant transformation is much more limited in the review, as compared to PRC2. I assume this reflects our deeper knowledge of the mechanisms of action and the transcriptional targets of PRC2, as compared to PRC1, in normal and malignant hematopoietic cells. There has been ongoing debate in the literature over the importance of the E3 Ub ligase catalytic activity of PRC1/Ring1B for its activity as a repressor of gene expression. Do we know whether the disruption of PRC1 activity, for example in cancers with BCOR mutations, causes transformations through loss or re-distribution of H2AK119ub epigenetic mark, or is it not directly linked to dysregulation in H2AK119ub, or do we not have these answers yet?
Response
Thank you for your question. The loss of BCOR induced reductions in H2AK119ub1 levels at a larger number of promoter regions of PRC1 targets, including Hoxa7, Hoxa9, and Cebpa (Tara S et al, Blood 2019;132:2470-83). We described this point on pages 6-7, lines 203-205.
“The loss of BCOR induced reductions in H2AK119ub1 levels at a larger number of promoter regions of PRC1 targets, including Hoxa7, Hoxa9, and Cebpa.”
- Do we know whether the regulation/repression of inflammatory genes by PRC2 discussed in the manuscript, including IL6, IL6Ra, S100As, etc, is a direct effect? Do we know if PRC2/Ezh2 is recruited to the regulatory regions of these genes, and Ezh2 loss directly results in loss/depletion of H3K27me3 at the bound site, leading to increased gene expression?
Response
Thank you for your question.
Regarding these inflammatory genes, we have not yet confirmed whether they are directly deregulated by Ezh2.
- Section on clinical trials of Ezh2 Inhibitors: Could you specify if the inhibitors show efficacy against epithelioid sarcoma and follicular lymphoma in general, or are the patients in these trials specifically pre-selected to include only those with Ezh2 gain-of-function mutations?
Response
Thank you for your suggestive comment. The reviewer’s request is quite reasonable.
Except for two clinical trials (NCT03456726, NCT03213665), most of them have recruited patients regardless of EZH2 mutations, presumably to ensure an adequate number of cases.
In this revised manuscript, we added the sentences on page 7, lines 254-255, as follows.
“Except for two clinical trials (NCT03456726, NCT03213665), most of them included patients without EZH2 mutations.”
We also revised the sentence in Table 2, as follows.
“R/R B-cell NHL harboring EZH2 mutations”
- One of the early studies reporting the role of BMI1 in the maintenance not only of normal HSCs but also of leukemic stem cells in mouse models can be further discussed (PMID: 12714970 DOI: 10.1038/nature01572).
Response
We really appreciate the reviewer’s comment on the function of BMI1 in the maintenance of leukemic stem cells in mouse models.
In the revised manuscript, we added the sentences on page 2, line 60, as follows.
“Bmi-1 is also required to maintain leukemic stem cells as well as normal HSCs (Nature. 2003;423:255-60).”
Minor questions and corrections
- Line 77, "Depletion of PCGF1 also led to myeloid-skewing [19] and repressed expression of HoxA family genes". Do you mean de-repressed? I believe hyperactivation of HoxA genes is usually linked to myeloid transformation.
Response
We really appreciate the kind reviewer’s comment.
In the revised manuscript, we changed the sentence on pages 2-3, line 77-78, as follows.
“Depletion of PCGF1 also led to myeloid-skewing [20] and de-repressed expression of HoxA family genes.”
- Lines 85-90, Two sentences here that are highly informative but lack citations. "Overexpression of PcG genes generally promotes tumorigenesis, partly through their ability to transcriptionally repress tumor suppressor genes, such as the CDKN2A locus (encoding p16INK4A and p14ARF), and developmental regulator genes. The expression of BMI1/PCGF4 has been correlated with disease progression and the prognosis of myelodysplastic syndrome (MDS) as well as the prognoses of acute myeloid leukemia (AML) and chronic myeloid leukemia." Please add some citations.
Response
Thank you for your essential comment.
In the revised manuscript, we additionally cited three papers.
We changed the sentence on page 3, lines 85-90, as follows.
“Overexpression of PcG genes generally promotes tumorigenesis, partly through their ability to transcriptionally repress tumor suppressor genes, such as the CDKN2A locus (encoding p16INK4A and p14ARF), and developmental regulator genes (Nat Rev Cancer. 2009;9:773-84). The expression of BMI1/PCGF4 has been correlated with disease progression and the prognosis of myelodysplastic syndrome (MDS) (Blood. 2006;107:305-8) as well as the prognoses of acute myeloid leukemia (AML) and chronic myeloid leukemia (Blood Cells Mol Dis. 2014; 53:194-8).”
- Similarly, Line 103 "EZH2 is overexpressed and/or amplified in prostate, breast, bladder, and colon cancers, and its expression is correlated with metastasis and poor prognosis." This is a very general statement, and some citations should be added. Also, can we have more specific information, how frequently EZH2 is dysregulated in these types of cancers? In majority of cases, frequently, occasionally? The sentence in its current form can be misinterpreted to mean that EZH2 activation is always seen in these cancers.
Response
Thank you for your constructive comment.
EZH2 is amplified in a significant portion of the patients with solid tumor, and the frequency depends on the type of cancer. It seems to be about 5-10% (Nat Rev Cancer. 2016;16(12):803-810). It is difficult to give an exact frequency of the overexpression because the definition of overexpression varies in each report. However, EZH2 is frequently overexpressed in these solid tumors.
In the revised manuscript, we additionally cited several papers that showed the data on frequencies of amplified EZH2 (Nat Rev Cancer. 2016;16(12):803-10), contribution to metastatic mechanisms (Nat Commun. 2015;6:6051), and its poor prognostic impact in solid tumor (Nature 2002; 419:624-9, Proc Natl Acad Sci U S A 2003; 100:11606-11).
We changed the sentence on page 5, lines 104-105, as follows.
“EZH2 is frequently overexpressed and/or amplified in prostate, breast, bladder, and colon cancers (Nat Rev Cancer. 2016;16:803-10), and its expression is correlated with metastasis (Nat Commun. 2015;6:6051) and poor prognosis (Nature. 2002; 419:624-9, Proc Natl Acad Sci U S A. 2003; 100:11606-11).”
- Line 149, "genes are remain", typo to be corrected.
Response
We really appreciate the kind reviewer’s comment.
In the revised manuscript, we changed the sentence on page 5, lines 150, as follows.
" genes are remain transcriptionally repressed even in EZH2-insufficient cells"
- The text on lines 160-165 is somewhat repetitive. It seems the same or similar information is stated several times. "The absence of the polycomb-group gene Ezh2 induces activation of a cohort of fetal-specific genes, including let-7 target genes, in adult BM HSPCs, leading to acquisition of fetal phenotypes by BM HSPCs, such as enhanced self-renewal activity and production of fetal-type lymphocytes [63]. Many of the fetal-specific let-7 target genes, including Lin28, Hmga2, and Igfbp3, are known as oncofetal genes, and are targeted by Ezh2-mediated H3K27me3 in adult BM HSPCs. Ezh2 loss results in their ectopic expression in adult BM HSPCs."
Response
We really appreciate the constructive reviewer’s comment.
To avoid repetitive description, we omitted some words on page 6, lines 161-163, as follows.
“The absence of the polycomb-group gene Ezh2 induces activation of a cohort of fetal-specific genes, including let-7 target genes, in adult BM HSPCs, leading to acquisition of fetal phenotypes by BM HSPCs, such as enhanced self-renewal activity and production of fetal-type lymphocytes [71]. Many of the fetal-specific let-7 target genes, including Lin28, Hmga2, and Igfbp3, are known as oncofetal genes, and are targeted by Ezh2-mediated H3K27me3 in adult BM HSPCs. Ezh2 loss results in their ectopic expression in adult BM HSPCs."
Reviewer 3 Report
Manuscript ijms-999878 submitted by Kaito and Iwama reviews the recent advances in functional aspects of PCR complex research in various blood cancers. The manuscript is well written in most part. Besides chemo resistance, does mutation in PRC genes also associated with radiotherapy?
Author Response
Dear Reviewer #3,
We really appreciated your comment.
Comment
Manuscript ijms-999878 submitted by Kaito and Iwama reviews the recent advances in functional aspects of PCR complex research in various blood cancers. The manuscript is well written in most part. Besides chemo resistance, does mutation in PRC genes also associated with radiotherapy?
Response
Thank for your essential questions.
Given that PRC genes mutations are frequently associated with poor prognosis, it might give radiotherapy resistance. Wu et al. showed that irradiation could decrease EZH2 expression using prostate cancer cell lines and its expression may associate with resistance to radiation therapy (Oncotarget. 2016; 7: 3440–3452). However, we were not able to find any evidences that PRC genes mutations correlate with resistance to radiotherapy in hematologic malignancies. We need to wait for other reports as to whether the PRC gene mutations lead to resistance to radiation therapy.
Round 2
Reviewer 1 Report
The authors have revised the manuscript and adequately addressed my concerns.